# Impact of the COVID-19 Pandemic on Carbapenem-Resistant *Klebsiella pneumoniae* in Northern Region, Brazil: A Shift Towards NDM Producers

**DOI:** 10.3390/antibiotics14090866

**Published:** 2025-08-28

**Authors:** Thalyta Braga Cazuza Barros, Yan Corrêa Rodrigues, Amália Raiana Fonseca Lobato, Lívia Maria Guimarães Dutra, Herald Souza dos Reis, Ana Judith Pires Garcia, Fernanda do Espírito Santo Sagica, Cintya de Oliveira Souza, Danielle Murici Brasiliense

**Affiliations:** 1Bacteriology and Mycology Deparment, Evandro Chagas Institute, Ministry of Health, Ananindeua 67030-000, Pará, Brazil; thalytacazuza@iec.gov.br (T.B.C.B.); amalialobato007@gmail.com (A.R.F.L.); liviadutra@iec.gov.br (L.M.G.D.); heraldreis@iec.gov.br (H.S.d.R.); anaquaresma@iec.gov.br (A.J.P.G.); cintyaoliveira@iec.gov.br (C.d.O.S.); 2Serra Talhada Academic Unit (UAST), Federal Rural University of Pernambuco (UFRPE), Av. Gregório Ferraz Nogueira, s/n, Serra Talhada 56909-535, Pernambuco, Brazil; yan.crodrigues@ufrpe.br; 3Central Laboratory of the State of Pará, Belém 66823-010, Pará, Brazil; 4Education Deparment, Evandro Chagas Institute, Ministry of Health, Ananindeua 67030-000, Pará, Brazil; fernandasagica@iec.gov.br

**Keywords:** *Klebsiella pneumoniae*, COVID-19, beta-lactam resistance, enterobacteriaceae, clinical epidemiology

## Abstract

**Background:** The global impact of the SARS-CoV-2 pandemic on antimicrobial resistance (AMR) patterns has been significant. In northern Brazil, Carbapenem-resistant *Klebsiella pneumoniae* (CRKP) are a major concern, with an observed shift from *Klebsiella pneumoniae* carbapenemase (KPC) to New Delhi metallo-β-lactamase (NDM) during the pandemic. **Methods:** This cross-sectional study analyzed 775 carbapenem-resistant *K. pneumoniae* isolates collected from 25 hospitals in the Brazilian Amazon Region (states of Pará and Acre) between 2018 and 2021. The isolates were tested for the presence of carbapenemase genes (*bla*_KPC_, *bla*_NDM_, *bla*_OXA-48_, *bla*_IMP_, *bla*_VIM_, *bla*_AIM_, *bla*_DIM_, *bla*_GIM_ and *bla*_SIM_). **Results:** Of the isolates analyzed, n = 653/775 (84%) were carbapenemase producers, with the most prevalent being *bla*_KPC_ n = 446/775 (57.5%) and *bla*_NDM_ n = 243/775 (31.4%). A significant increase in NDM producers was observed during the pandemic, rising from n = 1/250 (8.4%) pre-pandemic to n = 222/525 (42.3%) during the pandemic, while KPC producers declined from n = 172/250 (68.8%) to n = 274/525 (52.2%) (*p* < 0.001). Adult intensive care units (ICUs) were the primary source of isolates n = 357/775 (46%), with a notable increase in tracheal secretion and surveillance swab samples during the pandemic. Regression analysis confirmed a strong upward trend in the prevalence of *bla*_NDM_ (R^2^ = 0.778). **Conclusions:** The shift from KPC to NDM producers in northern Brazil highlights an evolving AMR landscape, partly driven by the pandemic. Strengthened infection control measures, antimicrobial stewardship and continuous surveillance are essential to mitigate the spread of NDM-producing *K. pneumoniae* in settings with limited resources.

## 1. Introduction

Antimicrobial resistance (AMR) is recognized as one of the most pressing global health threats, posing severe challenges to the treatment of infectious diseases worldwide [1]. The increasing prevalence of multidrug-resistant (MDR) microorganisms, particularly in healthcare settings, has significantly impacted clinical outcomes, resulting in higher rates of morbidity and mortality. Among these organisms, *Klebsiella pneumoniae* (Kp) has emerged as a leading cause of hospital-acquired infections, particularly in intensive care units (ICUs), due to its ability to acquire resistance mechanisms is a major concern [2,3]. In parallel, there has been a growing emergence of hypervirulent *Klebsiella pneumoniae* (hvKp) strains, which are responsible for community-acquired infections, often in otherwise healthy individuals. These microorganisms represent an emerging pathotype, exhibiting greater virulence compared to classical *K. pneumoniae* (cKp). The emergence of such strains has occurred, in part, through the acquisition of hvKp-specific virulence determinants by extensively drug-resistant cKp strains (XDR-cKp), leading to nosocomial infections caused by isolates with both hypervirulent and multidrug-resistant phenotypes [4].

The global health crisis driven by the SARS-CoV-2 pandemic has further exacerbated the AMR problem. During the pandemic, factors such as the inappropriate and excessive use of antibiotics in patients without bacterial co-infections, the overwhelming burden on healthcare systems, and the disruption of routine infection control measures, have contributed to an increase in AMR rates [5,6].

An example of this can be observed in studies conducted in Lebanon, which investigated the prevalence of inappropriate antibiotic use during the COVID-19 pandemic. The findings indicated that 40.2% of respondents reported using antibiotics during SARS-CoV-2 infection, and among them, 49% did so without a medical prescription. These data underscore that the majority of antibiotic consumption occurs outside the hospital setting, which facilitates improper use and highlights a significant lack of adequate public awareness [7].

Of particular concern is the increase in carbapenem-resistant *Klebsiella pneumoniae* (CRKP), a pathogen known for its capacity to harbor carbapenemase-encoding genes, which confer resistance to a wide range of antibiotics [8]. This scenario has also accelerated the spread of highly resistant strains with difficult-to-treat (DTR) and extensively drug resistant (XDR) phenotypes in healthcare settings, making the treatment of these infections more challenging given the reported limitations of antimicrobials [9,10].

In Brazil, the epidemiology of carbapenem resistance in *K. pneumoniae* in healthcare settings is predominantly associated with spread of high-risk clones (HRCs) containing mobile genetic elements and *Klebsiella pneumoniae* carbapenemase (KPC), a class A β-lactamase enzyme that has become endemic in the country [11,12,13,14].

However, in recent years, particularly following the onset of the SARS-CoV-2 pandemic, the prevalence of microorganisms producing New Delhi metallo-β-lactamase (NDM) has increased in healthcare settings in Brazil and other countries [15,16,17].

This is concerning not only because of the potential for the emergence and dissemination of other resistance mechanisms, but also because of resistance to newer β-lactamase inhibitors in metallo-β-lactamase producers, which makes treatment options even more limited.

The Northern region of Brazil has seen a concerning rise in CRKP infections but remains understudied in terms of detailed epidemiological data. This deficit is reflected in the scarcity of studies on AMR involving healthcare services in the Northern region, which presents several public health and basic sanitation infrastructure challenges in most states and municipalities in this region [18].

To clarify the epidemiology of CRKP in hospitals in the Northern region and assess the impact of the pandemic, this study investigated the prevalence and distribution of CRKP isolates collected from 25 hospitals in the Northern region of Brazil before and during the COVID-19 pandemic (2018–2021). By comparing temporal trends in carbapenemase production, this study sheds light on the evolving landscape of AMR in a region with limited resources.

## 2. Results

Of the 946 carbapenem-resistant *Klebsiella* spp. isolates received at the Evandro Chagas Institute, 171 were excluded due to duplication, contamination, or incorrect identification. These isolates did not belong to the *Klebsiella* genus. This left a total of 775 isolates for analysis. *K. pneumoniae* was the predominant species among the isolates, comprising over n = 754 (97%) of the total samples. The remaining isolates included *K. aerogenes* n = 13 (2%), *K. oxytoca* n = 5 (0.65%), and *K. ozaenae* n = 1 (0.13%). Of these isolates, 703 isolates were collected from health services in Pará State, and 72 isolates were collected from Acre State.

Analysis of the age groups demonstrated a predominance of isolates from elderly patients n = 301/775 (38.8%) and adults n = 288/775 (37.2%), with lower frequencies observed among children n = 83/775 (10.7%), young individuals n = 26/775 (3.3%) and newborns n = 12/775 (1.5%). Statistically significant variations were observed when comparing the pre-pandemic and pandemic periods in the adult group, which increased from 80 isolates (32%) in the pre-pandemic period to 209 isolates (39.8%) during the pandemic (*p* = 0.036). These and other values are better visualized in Table 1. Conversely, the pediatric group showed a significant decrease, from 45 isolates (18%) before the pandemic to 38 isolates (7.2%) during the pandemic (*p* = 0.001), as detailed in Table 1. The isolates were recovered from various clinical specimens, the majority of which were obtained from surveillance swabs n = 203/775 (26.2%), tracheal secretions n = 180/775 (23.2%), urine n = 170/775 (21.9%), and whole blood n = 121/775 (15.6%). Other clinical specimens of lower frequency can be seen in Figure 1.

Comparing the pre-pandemic period (2018–2019) with the pandemic period (2020–2021) revealed a significant increase in isolates obtained from tracheal secretions and surveillance swabs (Figure 2). The number of isolates obtained from tracheal secretions rose from 46 (18.4%) in the pre-pandemic period to 134 (25.5%) during the pandemic (*p* = 0.028). Similarly, the number of isolates obtained from surveillance swabs increased from 55 (22%) to 148 (28.2%) (*p* = 0.040) during the same period (Table 1). These findings reflect the increased use of invasive devices and enhanced surveillance measures, particularly among patients with SARS-CoV-2 infection, during the pandemic.

The majority of carbapenem-resistant *Klebsiella* spp. isolates were recovered from patients admitted to adult ICUs, accounting for n = 357/775 (46%) of the total isolates. This was followed by patients admitted to medical clinic wards, accounting for n = 225/775 (29%). The number of isolates from adult ICUs increased significantly during the pandemic, rising from 99 (40%) pre-pandemic to 267 (49%) during the pandemic period (*p* = 0.013). Isolates from pediatric ICUs and neonatal ICUs were less frequent, representing 4.4% and 1.8% of the total isolates, respectively. Notably, there was a significant decrease in neonatal ICU isolates during the pandemic period, falling from 3.2% pre-pandemic to 1.1% during the pandemic (*p* = 0.044) (Table 1).

Of the 775 isolates analyzed, n = 651/775 (84%) were confirmed to be carbapenemase producers, harboring one or more carbapenemases encoding genes. The most frequently detected gene was *bla*_KPC_ n = 446/775 (57.5%), followed by *bla*_NDM_ n = 243/775 (31.4%). Detection rates for other carbapenemase genes, such as *bla*_GES_ n = 14/775 (1.8%), *bla*_OXA-48_, and *bla*_IMP_, were very low, with only one isolate harboring each of the latter two genes. The *bla*_VIM_, *bla*_GIM_, *bla*_AIM_, *bla*_SIM_, and *bla*_DIM_ genes were not detected (Table 2). Although *K. pneumoniae* accounted for the highest frequency among carbapenemase-producing isolates, the *bla*_KPC_ and *bla*_NDM_ genes were also detected in non-classical *Klebsiella* species. For *bla*_KPC_, *K. aerogenes* presented 7 out of 13 isolates; *K. ozaenae* presented 1 out of 1; and *K. oxytoca* presented 2 out of 5. For *bla*_NDM_, *K. oxytoca* also presented 2 out of 5 isolates.

KPC-producing carbapenem-resistant *Klebsiella pneumoniae* (CRKP) isolates were detected in 23 of the 25 hospitals included in the study, with 18 hospitals located in Pará State and five in Acre State. Among these, n = 236/446 (52.9%) of KPC-positive patients were admitted to intensive care units (ICUs), underscoring the substantial prevalence of these resistant strains in critical care settings. The most frequent species was *K. pneumoniae*, accounting for n = 436/446 (97.7%) of the isolates. The *bla*_KPC_ gene was predominantly detected in isolates from tracheal secretions n = 119/446 (27%), urine n = 105/446 (24%), surveillance swabs n = 88/446 (20%), and blood n = 67/446 (15%). The most prevalent age groups among the individuals evaluated were the elderly, accounting for n = 165/446 (37.0%), and adults, with n = 160/446 (35.8%). Lower proportions were observed among children n = 43/446 (9.6%), young individuals n = 20/446 (4.4%), and neonates n = 3/446 (1.3%). The detection rate of KPC-positive CRKP was n = 67/105 (64.0%) in 2018, increasing to n = 107/145 (74.0%) in 2019. However, during the pandemic years, the detection rate declined, dropping to n = 76/138 (55.0%) in 2020 and further to n = 200/387 (52.0%) in 2021 (Table 2). When comparing the pre-pandemic period (2018–2019) with the pandemic period (2020–2021), the detection rate of KPC decreased significantly from 69% to 52% (*p* < 0.000). This analysis revealed a slight downward trend in the detection of KPC-producing CRKP isolates over the study period. A Pearson’s chi-square test indicated a significant difference (*p* < 0.000) in the frequency of KPC between 2019 and 2021 (Table 1). However, linear regression analysis demonstrated a weak correlation between the time period and the detection rate of the *bla*_KPC_ gene (R^2^ = 0.182), suggesting that temporal factors had a minimal influence on the decline of this carbapenemase (Figure 3).

On the other hand, NDM was the second most frequently detected carbapenemase, identified in n = 243/775 (31.4%) of the CRKP isolates. Isolates were identified in 17 out of the 25 hospitals included in the study, with 14 located in the capital of the state of Pará and 3 in municipalities in the countryside. No isolates were detected from the state of Acre. The predominant bacterial species was *K. pneumoniae,* accounting for n = 241/243 (99.1%) of the isolates. Among the *bla*_NDM_ producing isolates, n = 111/243 (45.7%) were obtained from patients admitted to Intensive Care Units (ICUs). Additionally, n = 78/243 (32.1%) of the isolates were recovered from clinical wards, n = 13/243 (5.3%) from pediatric ICUs, and n = 2/243 (0.8%) from neonatal ICUs.

The *bla*_NDM_ gene was primarily detected in surveillance swabs n = 95/243 (39%), blood n = 45/243 (19%), urine n = 42/243 (17%), and wound secretions n = 36/243 (15%). Regarding age distribution, adults and elderly individuals were the most affected groups, each representing n = 101/243 (41.5%) of cases. Lower frequencies were observed among children n = 22/243 (9%), neonates n = 6/243 (2.4%), and young individuals n = 5/243 (2%). Although the detection rate of *bla*_NDM_ was lower than that of *bla*_KPC_ in all years analyzed, a significant increase was observed over the study period. The detection rate rose from n = 3/105 (3.0%) in 2018 to n = 19/145 (13.0%) in 2019, followed by n = 54/138 (39.0%) in 2020, and n = 172/387 (44.0%) in 2021. A Pearson’s chi-square test revealed a highly significant association between these variables (*p* < 0.000), indicating a strong upward trend. Linear regression analysis showed a high correlation between the increase in *bla*_NDM_ detection and the progression of time (R^2^ = 0.778), confirming the marked rise in NDM prevalence, particularly during the COVID-19 pandemic (Figure 3).

## 3. Discussion

This study analyzed the prevalence and distribution of CRKP isolates in Northern Brazil before and during the COVID-19 pandemic, revealing a significant shift in the epidemiology of CRKP as a direct impact of the COVID-19 pandemic. The marked increase in NDM-producing CRKP isolates during the pandemic, rising from a low prevalence in 2018 and 2019 to a significant share by 2020 and 2021, was a key finding. This surge in NDM producers was statistically significant, indicated by strong linear regression analysis. Conversely, KPC-producing CRKP, while still prevalent, showed a decline in detection rates from 69% in the pre-pandemic years (2018–2019) to 52% during the pandemic (2020–2021). This suggests an evolving resistance landscape that has critical implications for patient outcomes and healthcare practices and aligns with global trends where emerging resistance mechanisms are reshaping the AMR profile in healthcare settings. Understanding these trends is essential for developing effective strategies to mitigate the impact of CRKP in a region already facing substantial public health and resource limitations.

The observed trends in Northern Brazil reflect shifts noted globally, where NDM-producing *K. pneumoniae* has gained prominence [19,20,21]. 

While KPC has historically dominated in Brazil and other regions, the rapid increase in NDM-producing strains during the COVID-19 pandemic marks a critical epidemiological change [22,23,24,25]. Similar trends have been reported in several countries in Europe, North America, and Latin America, where NDM detection increased significantly among ICU patients during the pandemic, and in China and India, where the spread of NDM in carbapenem-resistant Enterobacteriaceae affected both adult and pediatric populations [26,27,28,29,30].

Similar results were observed during a large surveillance study in Brazil that showed a 60% increase in resistance genes during the COVID-19 pandemic, especially *bla*_NDM_ in Enterobacterales [17].

The decrease in KPC detection over time, despite its still-high prevalence, aligns with previous reports of KPC being endemic in Brazil and in the North region [25,31,32]. The fact that KPC-producing isolates remained common in hospitals during the pandemic reflects the difficulty in eradicating these strains. However, the relatively stable levels of KPC producers compared to the sharp increase in NDM isolates suggest that NDM may overtake KPC as the dominant carbapenemase in *K. pneumoniae* clinical strains in the future.

The disproportionate impact of the pandemic on adult ICUs, as demonstrated by the significant increase in isolates from these wards, emphasizes the need for enhanced infection control and antimicrobial stewardship programs. Similar findings have been reported globally, where ICUs faced the brunt of the COVID-19 pandemic’s impact on AMR [33,34,35,36].

The surge in tracheal secretions and surveillance swabs further reflected the increased use of invasive devices and heightened infection control surveillance during COVID-19 patient care [37,38,39].

On the other hand, the increase in CRKP obtained from surveillance swabs is also relevant for epidemiological and infection control purposes, since colonized patients may develop infections with these microorganisms [40,41]. 

Pereira et al. (2021) [42] observed that among 70 patients with COVID-19 colonized by CRKP in ICUs, 14 developed infectious processes on average eight days after detection of colonization.

The increase in NDM-producing CRKP during the COVID-19 pandemic can be attributed to several factors. The pandemic led to the overuse of broad-spectrum antibiotics as empirical treatment or prevention for bacterial co-infections in COVID-19 patients, creating selective pressure that favored resistant organisms. This pattern has been observed in other studies where inappropriate antibiotic use during the pandemic contributed to the spread of resistant strains [43,44,45].

Interestingly, this study found a significant decrease in *K. pneumoniae* isolates from neonatal ICUs during the pandemic, with rates dropping from 3.2% pre-pandemic to 1.1% during the pandemic (*p* = 0.044). This reduction may be attributed to stricter infection control measures implemented to protect vulnerable neonatal populations, such as increased hygiene protocols, restricted visitor access, and heightened surveillance. Furthermore, the decrease may reflect shifts in healthcare resource allocation, where resources were redirected to adult ICUs managing COVID-19 cases. This redistribution could have reduced the use of invasive procedures in neonatal units, subsequently lowering opportunities for transmission. During the pandemic, neonatal units were likely prioritized for infection prevention to reduce risks of COVID-19 transmission, which may have indirectly lowered the transmission of other healthcare-associated infections, including *K. pneumoniae* [46,47,48].

The study revealed notably low detection rates for *bla*_GES_ n = 14/775 (1.8%), *bla*_OXA-48_ n = 1/775 (0.12%), and *bla*_IMP_ n = 1/775 (0.12%) among *Klebsiella* spp. isolates. This limited presence may reflect specific regional resistance patterns, where NDM and KPC are more prevalent, potentially due to transmission dynamics within healthcare settings. The low incidence of *bla*_OXA-48_ and *bla*_IMP_ among CRKP could indicate that these resistance mechanisms are not yet widely circulating in Northern Brazil, suggesting potential geographic or epidemiologic barriers to their spread [49].

The reliance on invasive medical devices such as ventilators and central lines, particularly in ICU settings, provided further opportunities for the transmission and colonization of resistant strains. In Northern Brazil, these challenges were exacerbated by limited healthcare resources and inconsistent infection control practices. The disruption of routine surveillance and preventive strategies facilitated the spread of NDM-producing CRKP, aligning with global observations of increased AMR during healthcare crises.

Epidemic high-risk clones of *K. pneumoniae*, such as ST258, ST11, ST15, and ST307, have played a leading role in the global spread of antimicrobial resistance. These clones are characterized by their ability to acquire and disseminate resistance genes rapidly, making them highly adaptable in diverse clinical environments [50,51].

The global dissemination of carbapenemase-producing *K. pneumoniae* can largely be attributed to these high-risk clones, which are associated with outbreaks in hospitals and have been linked to increased morbidity and mortality. In the context of CRKP, high-risk clones facilitate the spread of resistance mechanisms such as KPC, NDM, and OXA-48 carbapenemases. In Brazil, epidemic clones such as ST11 and ST437 have been identified in CRKP isolates, contributing to the dissemination of carbapenemase genes, including KPC and NDM. The presence of these clones exacerbates the public health challenge posed by AMR, as they contribute to sustained transmission within healthcare facilities [22,40,52,53].

The rise in NDM-producing CRKP has significant implications for clinical management and public health. Clinically, the increasing prevalence of NDM producers limits treatment options due to their resistance to most beta-lactam antibiotics, including newer beta-lactam/beta-lactamase inhibitor combinations. This often necessitates the use of less effective or more toxic alternatives, such as colistin, which is associated with nephrotoxicity and variable efficacy. In this context, the urgency of research and implementation of new antimicrobial strategies that expand the available therapeutic options is reinforced. One promising approach involves the use of bacterial metallophores, organic molecules produced by bacteria that are capable of sequestering metal ions essential for their growth and the expression of virulence factors. Overall, there is an increasing need to identify novel therapeutic pathways that are effective in combating multidrug-resistant microorganisms [54].

The COVID-19 pandemic revealed vulnerabilities in infection control that contributed to the spread of resistant pathogens. Addressing these gaps requires comprehensive and standardized infection control protocols, including rigorous hand hygiene, effective disinfection practices, and the proper use of Personal Protective Equipment (PPE). Healthcare facilities should implement strict patient isolation and cohorting measures to prevent cross-transmission. Regular training for healthcare staff on updated infection prevention practices and AMR stewardship is crucial for maintaining high compliance levels. Active surveillance to identify colonized or infected patients early can significantly curb transmission, supported by rapid diagnostic tools that enable timely intervention [55,56]. 

If current trends persist, NDM may surpass KPC as the dominant carbapenemase in Northern Brazil, complicating treatment regimens and straining healthcare resources. Strengthening antimicrobial stewardship to ensure judicious antibiotic use and investing in training for rapid diagnostic capabilities are key to managing this challenge. This study highlights the pressing need for further research into the factors driving the increased prevalence of NDM-producing CRKP. Investigations should focus on the genetic mechanisms facilitating NDM spread, the socio-economic impacts, and the long-term outcomes for patients with these infections. Continuous surveillance and comprehensive studies will inform public health strategies and guide policymaking to better address AMR challenges in the future.

## 4. Materials and Methods

### 4.1. Study Design and Selection of Bacterial Isolates

This is a cross-sectional, epidemiological, and analytical study that is part of the Antimicrobial Resistance Surveillance Study in North region, Brazil. It includes carbapenem-resistant (CR) *Klebsiella* spp. isolates obtained from various clinical specimens and surveillance swabs (axillary or rectal). The latter represents an important tool for hospital surveillance, aimed at screening patients colonized or infected with multidrug-resistant bacteria, thereby contributing directly to the prevention and control of nosocomial infections.

The patients were from adult, pediatric, and neonatal intensive care units (ICUs), as well as medical clinics, in 25 public and private healthcare facilities located in the Northern region of Brazil, including the states of Pará and Acre (Figure 4). Most of the healthcare facilities were in Belém city (N = 17), the capital of the state of Pará, while three were in other municipalities in the state of Pará. In the state of Acre, the five healthcare facilities included in this study were in the state capital, Rio Branco.

A total of 946 CR *Klebsiella* spp. isolates were received between January 2018 and December 2021. Only one bacterial isolate per patient was considered for the study. Duplicates, contaminated samples, and isolates not belonging to the *Klebsiella* genus were excluded (N = 171), leaving 775 *K. pneumoniae* isolates for analysis.

Preliminary bacterial identification and antimicrobial susceptibility testing (AST) were performed at the healthcare facilities. The isolates were then sent to Public Health Central Laboratories (LACENs) and subsequently to the Evandro Chagas Institute (IEC), where the laboratory analyses were conducted. These included seeding for thermal extraction of bacterial DNA and subsequent Polymerase Chain Reaction (PCR) and agarose gel electrophoresis.

### 4.2. Presence of Carbapenemases Encoding Genes

Bacterial isolates were subjected to DNA extraction following the protocol described by Vaneechoutte et al. (1995) [57] and subsequently tested for the presence of *bla*_KPC_, *bla*_NDM_, *bla*_OXA-48_, *bla*_IMP_, *bla*_VIM_ using conventional polymerase chain reaction (PCR). Primer sequences, master mix composition, and cycling conditions were previously described by Han et al. [58].

PCR was performed using the GoTaqTM G2 Flexi DNA Polymerase kit (Promega, Madison, WI, USA) under the following conditions: one initial cycle at 95 °C for 2 min, followed by 35 amplification cycles (95 °C for 30 s, 52 °C for 40 s, and 72 °C for 50 s), and a final extension at 72 °C for 10 min. In addition, other less frequently detected carbapenemase-encoding genes, such as *bla*_GES_, *bla*_GIM_, *bla*_AIM_, *bla*_SIM_, and *bla*_DIM_, were also screened using conventional PCR with primers designed according to Mendes et al. (2007) [59] and Han et al. (2020) [58]

PCR amplicons were visualized by agarose gel electrophoresis. A positive control was included in each run. The amplified products were visualized on 1% agarose gels stained with SafeDye nucleic acid stain (Cellco, São Paulo, Brazil).

### 4.3. Statistical Analysis

The data used in this study were obtained from the antimicrobial resistance surveillance database which is maintained by the Evandro Chagas Institute (IEC). Descriptive analyses were performed using Microsoft Excel 2010, and the results were presented in the form of graphs and tables. The data covered patient demographics (age groups), clinical specimen types, hospital wards, sample collection dates, and carbapenemase gene detection results. To assess the impact of the pandemic, samples collected in 2018 and 2019 were categorized as the ‘pre-pandemic period’, while those collected in 2020 and 2021 were categorized as the ‘pandemic period’. The study periods were stratified into quarters according to the monthly distribution within each year, in order to improve temporal organization and facilitate trend analysis.

Correlation analyses between variables were conducted using Pearson’s chi-square test and linear regression, with a significance level set at *p* ≤ 0.05. The 95% confidence interval was calculated using Statistical Package for the Social Sciences (SPSS) v20.0 and Biostat v5.3 software. Pearson’s chi-squared test was used to analyze the relationship between the pre-pandemic (2018–2019) and pandemic (2020–2021) periods of COVID-19 and various factors, including the type of clinical specimen, hospital ward, and detected carbapenemases. Additionally, the Linear Detection Rate (DR) was calculated by dividing the number of positive samples for KPC and NDM by the total number of samples received during the study period [60]. Regression analysis was performed using DR values to predict trends in carbapenemase detection increases or decreases over time.

## 5. Conclusions

This study provides a comprehensive analysis of CRKP in Northern Brazil, highlighting a significant shift from KPC to NDM-producing strains, particularly during the COVID-19 pandemic. The rise in NDM poses a serious challenge to treatment options, as these strains are resistant to newer antibiotics. The findings in the study also stress the need for strengthened infection control, antimicrobial stewardship, and ongoing surveillance to curb the spread of NDM-producing pathogens, especially in ICUs. Continued monitoring is crucial to managing future threats of antimicrobial resistance in the region.

## Figures and Tables

**Figure 1 antibiotics-14-00866-f001:**
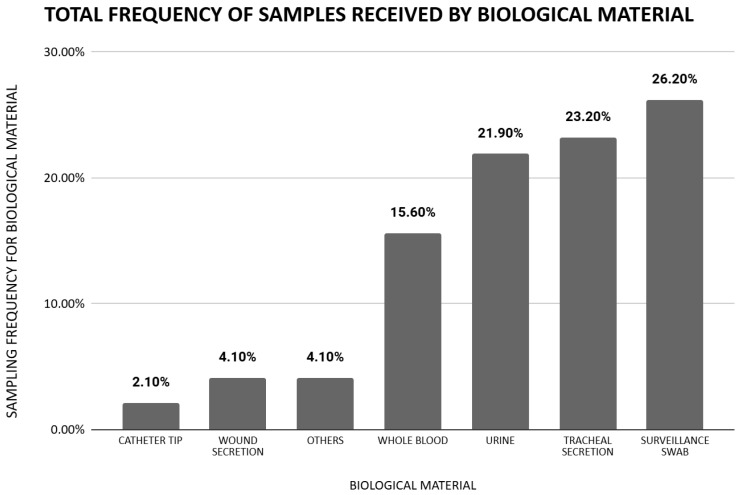
Frequency of carbapenem-resistant *Klebsiella* spp. isolates by clinical specimen.

**Figure 2 antibiotics-14-00866-f002:**
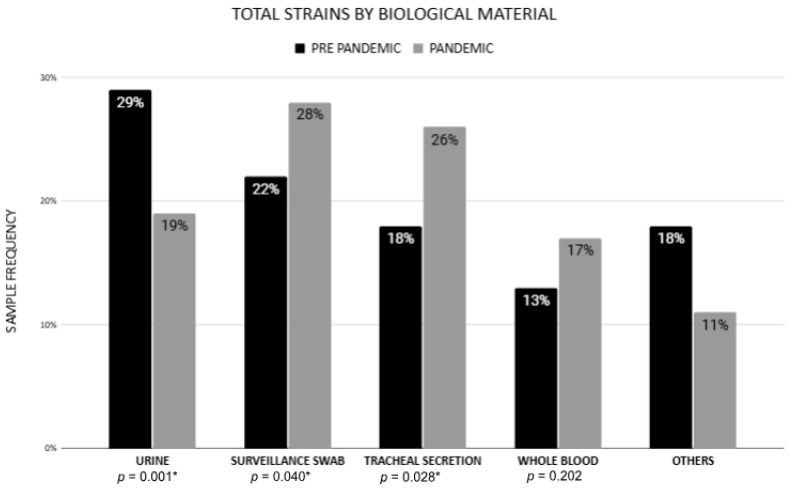
Frequency of Carbapenem-Resistant *Klebsiella* spp. Isolates per Clinical Sample: Pre-Pandemic vs. Pandemic Periods. (*) Statistically significant results.

**Figure 3 antibiotics-14-00866-f003:**
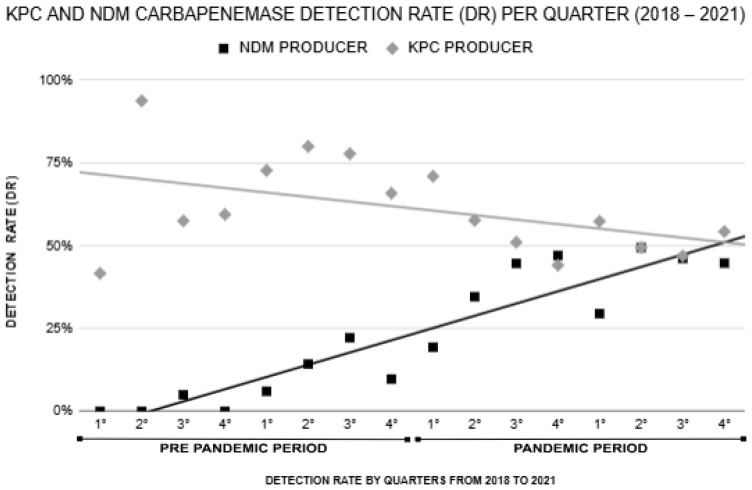
Time trend pattern for *bla*_NDM_ and *bla*_KPC_ detection rate by quarter from 2018 to 2021.

**Figure 4 antibiotics-14-00866-f004:**
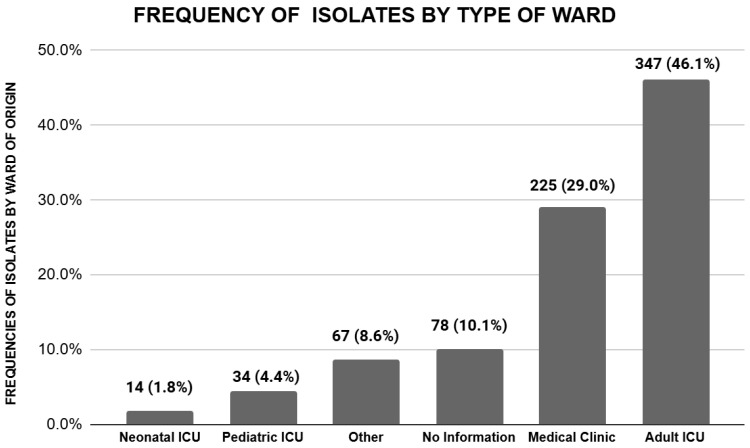
Total frequency of *Klebsiella* spp. isolates by ward of origin.

**Table 1 antibiotics-14-00866-t001:** Statistical analysis of correlation between the pre pandemic and pandemic period and variables.

	Pre-Pandemic Period N (%)	Pandemic Period N (%)	*p* Value
Correlation—Carbapenemases			
Carbapenemase producer	193 (77.2%)	451 (85.9%)	0.003 *
KPC producer	172 (68.8%)	274 (52.2%)	0.000 *
NDM producer	21 (8.4%)	222 (42.3%)	0.000 *
Correlation—Type of hospitalization ward			
Adult UCI	99 (39.6%)	258 (49.1%)	0.013 *
Pediatric UCI	16 (6.4%)	18 (3.4%)	0.059
Neonatal UCI	8 (3.2%)	6 (1.1%)	0.044 *
Medical clinics	67 (26.8%)	158 (30.1%)	0.345
Correlation—Clinical sample			
Whole blood	33 (13.2%)	88 (16.8%)	0.202
Catheter tip	5 (2%)	11 (2.1%)	0.931
Tracheal secretion	46 (18.4%)	134 (25.5%)	0.028 *
Urine	72 (28.8%)	98 (18.7%)	0.001 *
Surveillance swab	55 (22%)	148 (28.2%)	0.040 *
Wound secretion	9 (3.6%)	23 (4.4%)	0.609
Correlation—Age range			
NB (Up to 1 month)	7 (2.8%)	5 (1%)	0.051
Child (0–12)	45 (18%)	38 (7.2%)	0.001 *
Young (13–18)	10 (4%)	16 (3%)	0.491
Adult (19–59)	80 (32%)	209 (39.8%)	0.036 *
Elderly (+60)	105 (42%)	196 (37.3%)	0.213

(*) Statistically significant results.

**Table 2 antibiotics-14-00866-t002:** Detection rate of carbapenemase genes in *Klebsiella* spp. from 2018 to 2021.

Year of Sample Collection	Carbapenemase Genes Detected					
	*bla* _KPC_	*bla* _NDM_	*bla* _OXA-48_	*bla* _IMP_	*bla* _VIM_	Coproduction *bla*_KPC_ + *bla*_NDM_
2018	64% (67/105)	3% (3/105)	0% (0/0)	0% (0/0)	0% (0/0)	0% (0/0)
2019	74% (107/145)	13% (19/145)	0% (0/0)	0% (0/0)	0% (0/0)	1.3% (2/145)
2020	55% (76/138)	39% (54/138)	0.7% (1/138)	0% (0/0)	0% (0/0)	6.5% (9/138)
2021	52% (200/387)	44% (172/387)	0% (0/0)	0.2% (1/387)	0% (0/0)	10% (39/387)

## Data Availability

The data used in this study can be made available upon request to the corresponding author, considering the sensitivity of patient information and the confidentiality requirements of the research.

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
