# Peer review of "Impact of the COVID-19 Pandemic on Carbapenem-Resistant Klebsiella pneumoniae in Northern Region, Brazil: A Shift Towards NDM Producers"

_antibiotics, 2025, doi:10.3390/antibiotics14090866_

Round 1
Reviewer 1 Report
Comments and Suggestions for Authors
Dear Authors,
Your original research article entitled "Impact of the COVID-19 Pandemic on Carbapenem-Resistant Klebsiella pneumoniae in Northern Region, Brazil: A Shift Towards NDM Producers" has been carefully reviewed.
This article deserves attention since it highlights on a very important topic related to the impact of the last worldwide pandemic "COVID-19" on Carbapenem Resistant Klebsiella pneumoniae (CRKP) and the possible shift to New Delhi Metallo-β-lactamase (NDM) resistance ability, in the Northern Region of Brazil (A large Country in Latin America).
The article is well written in English language, well presented and well designed, Figures and Tables are very clear for readers.
Kindly find below my list of comments regarding the present work:
01- In the whole manuscript the name of bacterial species should be in italic. Example in line 14 (Klebsiella pneumoniae)
02- In the Abstract section, (CR Kp) is not correct the term is (CRKP) and it stands for Carbapenem Resistant Klebsiella pneumoniae and not for Klebsiella pneumoniae
carbapenemase producers.
03- In the list of keywords, you are kindly invited to put "Klebsiella pneumoniae".
04- When you talked about the excessive and misuse of antibiotics during the pandemic, you are invited to use the following article as reference for this point:
-- Antibiotic Misuse during the COVID-19 Pandemic in Lebanon: A Cross-Sectional Study.
05- In the whole manuscript, just for the first time you used the full name of the bacterium "Klebsiella pneumoniae" then you can replace it by its abbreviation "K. pneumoniae" or too simply "Kp".
06- In the Introduction section, between Line 64 and 73, there is duplicated information in the two paragraph, you are kindly invited to remove duplicated information.
07- In the Introduction section, I suggest to talk about the worldwide increase pathogenicity of Klebsiella pneumoniae mainly you can talk about what we called "Hypervirulent Klebsiella pneumoniae", you can use the following paper as reference for this point:
-- Hypervirulent Klebsiella pneumoniae.
08- Can you please be more persuasive when you talked about surveillance swabs (How and Why it was collected?)
09- In Graph one, can you please put it in a descending order?
10- Why you results are limited to 2021? Why you don't have data from the post-pandemic phase until 2024 for example?
11- Can you explain how your years were divided into 4 quartiles? Based on seasons? or based on months?
12- On Graph 3, it is good to divide it into pre-pandemic and pandemic period.
13- When talking in the discussion, I think it will be important to mention the importance of fighting AMR by creating new families of antibiotics targeting new structures in bacteria for example targeting bacterial metallophores or the trojan horse technic. You can use the following two papers as references for this point:
-- Towards new antibiotics classes targeting bacterial metallophores
-- Trojan Horse Antibiotics—A Novel Way to Circumvent Gram-Negative Bacterial Resistance?
14- You can remove the section 6 "Abbreviations" from the article.
Best Regards,
Author Response
Comments 1, 2, 3, 5, 6, 8, 9, 12, 13, and 14:
All the points raised were carefully reviewed and addressed in the revised manuscript. We are grateful for the reviewer’s suggestions, which significantly contributed to improving the quality of the work, both structurally and conceptually, allowing for the inclusion of relevant discussions.
Comment 10:
As this study was concluded in 2022, the authors do not have access to data beyond the years covered in the analysis.
Comment 11:
Given that samples were received in nearly all months throughout the study period, data were stratified into quarters based on monthly distribution. This approach aimed to enhance the temporal organization and facilitate the analysis of data trends over time. This information has been added to the "Statistical Analysis" subsection of the Materials and Methods.
Reviewer 2 Report
Comments and Suggestions for Authors
The manuscript presents a sound methodological approach and addresses a topic of significant epidemiological relevance, particularly regarding carbapenemase-producing Klebsiella strains. It is worth noting that KPC-producing strains have been documented in South America for over a decade, with sustained epidemiological importance both before and after the COVID-19 pandemic.
However, several key points require clarification to enhance the scientific robustness of the study:
Presence of multiple carbapenemase genes in a single strain: Table 2 reports the detection of individual resistance genes by year; however, it does not indicate whether any of the strains carried more than one carbapenemase gene simultaneously. This detail is essential for understanding the complexity of the resistance mechanisms involved.
Lack of information on non-classical Klebsiella species: The manuscript does not report the presence of carbapenemase genes in K. aerogenes, K. oxytoca, and K. ozaenae, which account for 19 out of the 775 strains analyzed. Given their frequency, the inclusion of detected genes in these species is recommended.
Insufficient methodological detail regarding primer design and gene selection: The methods section does not specify the databases or references used to design the primers, nor does it explain the criteria for selecting the genes included in the analysis.
Potential value of phylogenetic analysis: The discussion does not consider the possibility of conducting a phylogenetic analysis of the isolates. Incorporating such analysis would strengthen the manuscript, especially in light of existing reports from South America documenting the circulation of KPC-2- and NDM-1-producing Enterobacteriaceae, which share phylogenetic characteristics relevant for genomic surveillance.
Comments on the Quality of English Language
Es correcto
Author Response
Comment 1: Presence of multiple carbapenemase genes in a single strain: Table 2 reports the detection of individual resistance genes per year; however, it does not indicate whether any of the strains carried more than one carbapenemase gene simultaneously. This detail is essential to understand the complexity of the resistance mechanisms involved.
Response: In response to the reviewer’s suggestion, an additional column was added to Table 2 showing the frequency of strains co-producing KPC and NDM. This modification allows a clearer understanding of the simultaneous presence of multiple carbapenemase genes within a single strain.
Comment 2: Lack of information on non-classical Klebsiella species: The manuscript does not report the presence of carbapenemase genes in K. aerogenes, K. oxytoca, and K. ozaenae, which represent 19 of the 775 strains analyzed. Given the frequency of these species, it is recommended to include the detected genes for these species.
Response: The Results section has been updated to include information on KPC and NDM production in non-classical Klebsiella species. This inclusion ensures that the analysis fully reflects the diversity of species present in the study.
Comment 3: Insufficient methodological details regarding primer design and gene selection: The Methods section does not specify the databases or references used to design the primers, nor explain the criteria for selecting the genes included in the analysis.
Response: The Materials and Methods section has been revised to include detailed information on reagents, PCR and electrophoresis protocols, as well as references used for primer design and criteria applied for the selection of the genes analyzed.
Comment 4: Potential value of phylogenetic analysis: The discussion does not consider the possibility of conducting a phylogenetic analysis of the isolates. Incorporating such analysis would strengthen the manuscript, particularly in light of reports from South America documenting the circulation of KPC-2- and NDM-1-producing Enterobacteriaceae sharing relevant phylogenetic characteristics for genomic surveillance.
Response: We acknowledge the relevance of phylogenetic analysis to enhance the discussion of our findings. However, due to financial constraints and the large number of isolates, performing such analysis was not feasible for this study. Nevertheless, we emphasize that the presented findings significantly contribute to understanding the dissemination of KPC and NDM genes in the studied region.
Reviewer 3 Report
Comments and Suggestions for Authors
The work by Cazuza Barros analysed 775 carbapenem-esistant K. pneumoniae isolates collected from 25 hospitals from Brasil between 2018 and 2021. This work is of utility and gives an adequate panorama in Brazil, and presents and interesting finding in the shift from KPC to NDM. The manuscript is well presented and disccussed. I just marked some minor comments in the pdf document.

Author Response
Dear Reviewer,
We sincerely thank you for your valuable and insightful comments. The authors have carefully reviewed the annotated PDF and implemented all the suggested corrections. Your contributions were essential in improving the quality of the manuscript.
Round 2
Reviewer 1 Report
Comments and Suggestions for Authors
Dear Authors,
The revised version of your original research article entitled "Impact of the COVID-19 Pandemic on Carbapenem-Resistant Klebsiella pneumoniae in Northern Region, Brazil: A Shift Towards NDM Producers" has been carefully reviewed.
The paper is more suitable for publication in its present form, thanks to the modifications you made,
Best Regards,